# Gray Matter Changes in Juvenile Myoclonic Epilepsy. A Voxel-Wise Meta-Analysis

**DOI:** 10.3390/medicina57111136

**Published:** 2021-10-20

**Authors:** Dimitrios Kazis, Foivos Petridis, Symela Chatzikonstantinou, Eleni Karantali, Rabee Jamali, Rumana Chowdhury, Raluca Duta, Alina-Costina Luca, Alin Ciobica, Ioannis Mavroudis

**Affiliations:** 1Third Department of Neurology, Aristotle University of Thessaloniki, 541 24 Thessaloniki, Greece; dimitrios.kazis@gmail.com (D.K.); f_petridis83@yahoo.gr (F.P.); melina.chatzik@gmail.com (S.C.); lena.kar@outlook.com (E.K.); i.mavroudis@nhs.net (I.M.); 2Department of Neurology, Leeds Teaching Hospitals, NHS Trust, Leeds LS2 9JT, UK; rabee.jamali@nhs.net (R.J.); rumana.chowdhury@nhs.net (R.C.); 3Department of Biology, Faculty of Biology, “Alexandru Ioan Cuza” University of Iasi, 700505 Iasi, Romania; duta.raluca112@gmail.com; 4Department of Pediatric Cardiology, Faculty of Medicine, “Grigore T. Popa” University of Medicine and Pharmacy, 700505 Iasi, Romania

**Keywords:** VBM, voxel-wise meta-analysis, juvenile myoclonic epilepsy, gray matter changes

## Abstract

*Background and Objectives*. Juvenile myoclonic epilepsy (JME) is an idiopathic generalized epileptic syndrome, with a genetic basis clinically identified by myoclonic jerks of the upper limbs upon awaking, generalized tonic-clonic seizures and less frequent absences. Although the brain magnetic resonance imaging (MRI) is by definition normal, computer-based Voxel-Based morphometry studies have shown a number of volumetric changes in patients with juvenile myoclonic epilepsy. Thus, the aim of the present Voxel-Wise Meta-Analysis was to determine the most consistent regional differences of gray matter volume between JME patients and healthy controls. *Materials and Methods.* The initial search returned 31 studies. After excluding reviews and studies without control groups or without detailed peak coordinates, 12 studies were finally included in the present meta-analysis. The total number of JME patients was 325, and that of healthy controls was 357. *Results.* Our study showed a statistically significant increase of the gray matter in the left median cingulate/paracingulate gyri, the right superior frontal gyrus, the left precentral gyrus, the right supplementary motor area and left supplementary motor area. It also showed a decrease in the gray matter volume in the left thalamus, and in the left insula. *Conclusions.* Our findings could be related to the functional deficits and changes described by previous studies in juvenile myoclonic epilepsy. In this way, the volumetric changes found in the present study could be related to the impaired frontal lobe functions, the emotional dysfunction and impaired pain empathy, and to the disrupted functional connectivity of supplementary motor areas described in JME. It additionally shows changes in the volume of the left thalamus, supporting the theory of thalamocortical pathways being involved in the pathogenesis of juvenile myoclonic epilepsy.

## 1. Introduction

Juvenile myoclonic epilepsy (JME) is a common syndrome of idiopathic generalized epilepsy, with a prevalence of 4–10% among all epileptic syndromes [1]. The onset is often in the middle of the second decade of life. JME has a high genetic basis. It is clinically characterized by myoclonic jerks of the upper limbs upon waking, generalized tonic-clonic seizures and absence seizures. The seizures usually follow a circadian rhythm. Common triggering factors for epilepsy, such as sleep deprivation, stress, fatigue, alcohol intake, or complex cognitive tasks can precipitate an episode in JME.

JME is characterized by 3–6 hertz (Hz) generalized spike-wave or polyspike-wave discharges on the electroencephalogram (EEG), with frontocentral accentuation [2].

Neuroimaging is by definition normal in patients with JME [2]; however, quantitative studies using voxel-based morphometry (VBM) have shown a number of gray matter abnormalities in comparison to healthy controls. Cao et al. conducted a meta-analysis of VBM studies on gray matter volume changes in JME in 2013. They included seven studies with a total of 211 JME patients and 241 healthy controls and found an increased gray matter volume in the bilateral medial frontal gyrus and anterior cingulate gyrus, while the thalamus showed a decreased volume bilaterally. They suggested that their findings were indicative of thalamocortical circuitry involvement in the pathophysiology of JME [3].

Since then, the number of voxel-based morphometric studies have increased significantly. In the present study, we did a voxel-based meta-analysis to determine the most consistent regional differences of gray matter volume between JME and healthy controls, including the studies used in the study by Cao et al. [3] and five additional studies were not included in their meta-analysis and were published up to 2021.

## 2. Materials and Methods

For the purpose of the present study, we conducted a search in the online databases of PubMed, Cochrane library, Web of Science and BrainMap using the keywords [“juvenile myoclonic epilepsy” OR “JME”] AND [“voxel-based”, OR “morphometry” OR “VBM”]. We only included studies that compared the gray matter changes in patients with juvenile myoclonic epilepsy to healthy controls. All the studies reported the results in the Montreal Neurological Institute (MNI) or Talairach space.

We did not include studies without detailed coordinates or without control groups. Two independent researchers extracted the number of participants and demographics, as well as the peak coordinates which had been used to prepare files in .txt format as per Seed-based d Mapping (SDM) software guidelines [4]. We used version 6.21 of the software in a 64 bit Windows PC. The full width at half-maximum was set at 20 mm, and the statistical threshold was *p* < 0.005. These values have been shown by previous studies to have the best control for false positive rates and are able to optimize the balance between sensitivity and specificity [5]. We also performed a jacknife sensitivity analysis by omitting one study at a time and repeating the mean analysis, and a heterogeneity analysis using a random-effects model with Q statistics [5,6].

### Risk of Bias

We evaluated the risk of bias for each of the studies using the Cochrane Risk of Bias tool. The publication bias was examined by Egger’s test and a visual inspection of the funnel plots for each comparison.

## 3. Results

The initial search returned 31 studies. After excluding reviews and studies without control groups or without detailed peak coordinates, 12 studies were finally included in the present meta-analysis [7,8]. The total number of JME patients was 325, and that of healthy controls was 357. This number of studies reported the coordinates in MNI space and in Talairach space (Figure 1).

### 3.1. Details of the Included Studies

Tae et al. investigated the structural brain abnormalities with JME in 19 patients and 19 age- and gender-matched healthy controls. They reported complex abnormalities in the frontal lobe and hippocampus and a decreased gray matter volume in the prefrontal region [9]. Kim et al. conducted a voxel-based morphometry analysis on the structural changes of the thalamus in 50 JME patients and 50 healthy controls using a 3T MRI scanner. They found a significant regional gray matter decrease in anterior-medial thalami bilaterally in JME patients [7]. A different study from the same group of 25 JME patients and 44 age- and gender-matched controls found increases in the superior mesiofrontal region bilaterally and a reduction of the gray matter in thalami bilaterally [10]. De Ajaujo et al. did a VBM study on 38 JME patients, 30 controls, and 16 JME patients with personality disorders. They used a 1.5T MRI scanner and reported a reduction of the gray matter volume in both thalami in JME patients compared to controls, a significant decrease in the left and right insula, and in the left and right cerebellar hemispheres. They also found an increased gray matter volume in JME in the right medial frontal gyrus. Between groups, the analysis showed certain differences between the two JME groups. They reported a significant reduction of the gray matter volume in the right thalamus in JME patients with personality disorders (PD) compared to JME without PD and an increase of the gray matter volume in the left and right middle frontal gyrus and in the right orbitofrontal cortex in the JME PD group [11]. Mory et al. also identified areas of atrophy in the anterior thalamus in 21 patients with JME compared to 20 age- and gender-matched healthy controls [12]. O’Muircheartaigh et al. performed a VBM study on 28 JME patients and a large sample of healthy controls, and they described a reduction in gray matter volume in the supplementary motor area and the posterior cingulate cortex in the JME group [13]. Swarzt et al. investigated the brain volumes in 17 patients with juvenile myoclonic epilepsy and 17 controls. They did not find any statistically significant difference in the volume of the gray matter between the two groups in their study [14] Liu et al. assessed the gray and white matter volume using a diffusion tensor tractography analysis and voxel-based morphometry in 25 patients with idiopathic generalized epilepsy, 15 of whom had JME and a number of controls. They reported no gray matter volume differences between the control and JME patients’ groups [15]. Furthermore, Roebling et al. reported no statistically significant differences in the gray matter volume in a VBM study on 19 JME patients and 20 age-, gender- and education-matched controls [16]. Betting et al., on the other hand, investigated the gray matter differences in 44 JME patients and 47 controls using voxel-based morphometry. They reported an increased gray matter volume in the frontobasal region in the JME group [17]. Finally, Woermann et al. performed a voxel-based morphometric study on 20 patients with JME and 30 controls. Their study revealed a significant increase in cortical gray matter in the medial frontal lobes of JME patients [18].

### 3.2. Gray Matter Changes in JME

#### 3.2.1. GM Increases

Our meta-analysis showed increased gray matter in JME patients compared to controls in the left median cingulate/paracingulate gyri (Brodmann area 23, SMD-Z: 1.404, *p* = 0.0002, Voxels: 820, MNI Coordinates: −4, −6, 42), the right superior frontal gyrus (Brodmann area 10, SMD-Z: 1.446, *p* = 0.0002, Voxels: 530, MNI coordinates: 20, 60, 18), the left precentral gyrus (Brodmann area 6, SMD-Z: 1.093, *p* = 0.0017, Voxels: 186, MNI coordinates: −20, −20, 66), the right supplementary motor area (Brodmann area 6, SDM-Z: 1.135, *p* = 0.0013, Voxels: 152, MNI Coordinates: 8, 10, 60) and left supplementary motor area (Brodmann area 6, SDM-Z: 1.139, *p* = 0.0013, Voxels: 72, MNI Coordinates: −8, 12, 68) (Figure 2A,B).

#### 3.2.2. GM Decreases

JME patients exhibited a decrease of gray matter volume in the left thalamus (SDM-Z: −1.1875, *p* = 0.00001, Voxels: 970, MNI Coordinates: −12, −6, 8), and in the left insula (Brodmann area 48, SDM-Z: −1.372, *p* = 0.0008, Voxels: 553, MNI Coordinates: 42, 2, 6) (Figure 2).

### 3.3. Heterogeneity, Publication Bias and Influence Analysis

The visual inspection of the funnel plots and Egger’s tests did not show significant evidence of publication bias (Figure 3).

A jackknife analysis was performed by omitting one study each time and repeating the mean analysis and showed that both the positive and negative peaks were robust, as they were replicable.

## 4. Discussion

Although the ILAE definition of JME states that brain imaging is normal [14,19] both histopathologic studies and voxel-based morphometric studies show evidence for structural brain changes in this population [20], with volumetric changes in the medial prefrontal cortex and thalamus [11,18].

Our meta-analysis identified an increased gray matter volume in the frontal brain regions. This regional difference is in keeping with studies using other modalities to show differences in these brain regions in JME compared to controls. For example, earlier studies had linked JME with a specific personality profile, with social immaturity, disinhibition, and a lack of endurance, similar to behavioral changes seen in patients with frontal lobe injuries [21]. The frontal lobe plays a crucial role in working and prospective memory, and in executive functions. Swartz et al., in their study on visual working memory using nine patients with JME, 15 patients with frontal lobe epilepsy (FLE) and 15 controls, reported an impairment of the working memory in the JME group compared to the controls and FLE patients [22]. The same group performed another study with 18FDG-PET to evaluate the visual working memory in nine JME patients and 14 controls and reported the poor performance of the JME patients on a working memory task and a decreased 18FDG uptake in the ventral premotor cortex, caudate, the dorsolateral prefrontal cortex bilaterally, and the left premotor area [23]. Devinsky et al., Pascalicchio et al., and Kim et al., [24,25,26] also reported an impaired working memory in JME patients, while other studies, however, showed a comparable performance in working memory tasks in JME patients and controls [27,28,29,30]. Studies on executive functions have also reported controversial results, while several studies found an impairment of the mental flexibility and the response inhibition and fluency, and others failed to find significant differences between JME patients and healthy controls [7,13,16,24,25,28]. In addition to frontal lobe functional impairment, patients with JME exhibit emotional dysfunction with reduced pain empathy, associated with a reduced neural activation of the anterior insula and the anterior cingulate cortex [31]. 

Previous evidence had shown a reduced functional connectivity in the supplementary motor areas in JME [32,33]. Our findings could therefore represent compensatory changes, though we acknowledge that volumetric differences do not map directly onto functional differences.

Our meta-analysis also found a reduced gray matter volume in the thalamus in JME. 

Significant alterations of the thalamic nuclei volumes and intrinsic thalamic networks have been described in JME patients [26], whilst SPECT studies have found a reduction of the regional cerebral blood flow in the bilateral thalamus [18]. Jiang et al. reported a decreased functional connectivity of thalamic regions with the superior frontal gyrus and an enhanced functional connectivity with the supplementary motor area, and they suggested that the posterior thalamus had been linked with the epileptic activity in JME [34]. 

Cao et al. conducted a meta-analysis of seven VBM studies in JME and reported an increase in the gray matter of the right medial frontal gyrus, the left medial frontal gyrus, and a reduction of gray matter in the thalamus bilaterally [3]. In the present voxel-wise meta-analysis, we found evidence of an increased gray matter volume in the left median cingulate/paracingulate gyri, the right superior frontal gyrus, the left precentral gyrus, and the supplementary motor areas bilaterally. Additional decreases of the volume of the gray matter were found in the left thalamus and the left insula. The present study therefore both corroborates and extends the earlier findings by Cao et al. [3] and gives new insight into the volumetric changes in JME.

Thus, the present study provides a comprehensive and robust assessment of volumetric changes in the gray matter of patients with JME compared to healthy controls. These findings are in keeping with regional pathological and functional differences described by other studies.

Still, our study has a few limitations. Although the number of studies is considerably bigger than previous meta-analyses on the same matter, an even bigger number of studies would make the results more robust and reliable. We limited our search to studies written only in English, which might have excluded a number of studies written in other languages. This study is a meta-analysis based on the data provided by other studies and not on raw data that could give more accurate results.

## 5. Conclusions

Our findings could be related to the functional deficits and changes described by previous studies in juvenile myoclonic epilepsy. In this way, the volumetric changes found in the present study could be related to the impaired frontal lobe functions, the emotional dysfunction and impaired pain empathy, and to the disrupted functional connectivity of the supplementary motor areas described in JME. Additionally, it shows changes in the volume of the left thalamus, supporting the theory of thalamocortical pathways being involved in the pathogenesis of juvenile myoclonic epilepsy.

## Figures and Tables

**Figure 1 medicina-57-01136-f001:**
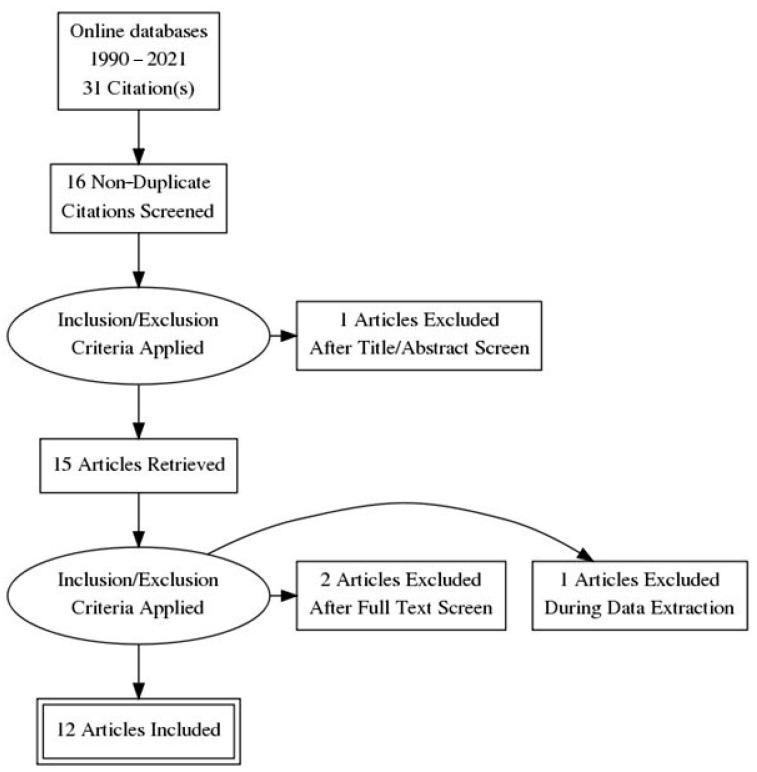
PRISMA flow diagram of the studies selection process.

**Figure 2 medicina-57-01136-f002:**
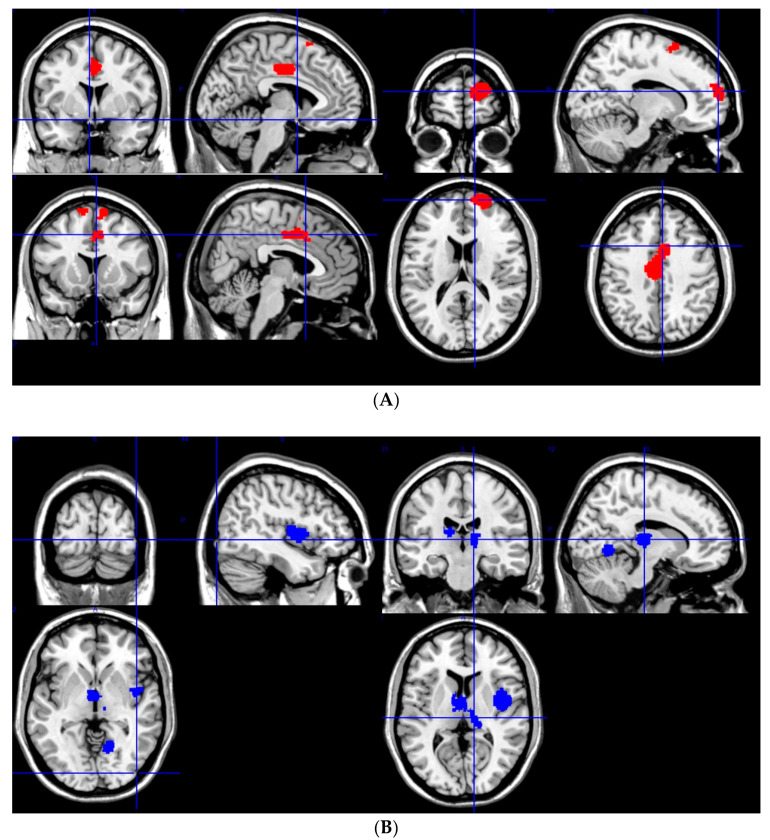
(**A**) Areas with increased gray matter volume in JME patients and (**B**) areas with decreased gray matter volume in JME patients compared to controls.

**Figure 3 medicina-57-01136-f003:**
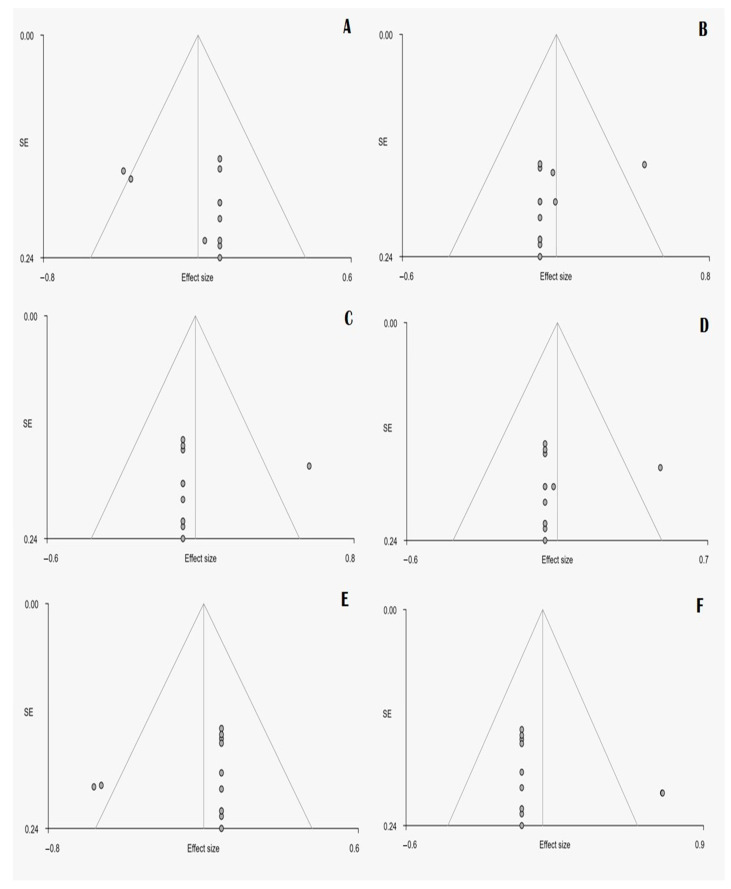
Funnel plot diagrams on all the areas that showed statistically significant differences between JME patients and controls. (**A**) Thalamus, (**B**) Left median cingulate cortex, (**C**) Left precentral gyrus, (**D**) Left supplementary motor area, (**E**) Right Insula, (**F**) Right superior frontal gyrus.

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
