# Peer review of "Gray Matter Changes in Juvenile Myoclonic Epilepsy. A Voxel-Wise Meta-Analysis"

_medicina, 2021, doi:10.3390/medicina57111136_

Round 1

Reviewer 1 Report

Thank you for the opportunity for reviewing this manuscript.

The authors did a voxel-based meta-analysis of 12 VBM studies to determine the most consistent regional differences of gray matter volume between juvenile myoclonic epilepsy (JME) and heathy controls adding 5 studies published from 2006 to 2021 to the previous study done by Cao in 2012. The authors found increased gray matter volume in frontal brain regions (the left median cingulate/paracingulate gyri, the right superior frontal gyrus, the left precentral gyrus, the bilateral supplementary motor areas) and reduced gray matter volume in the left thalamus and right insula in JME. The authors speculated these findings are related to the impaired frontal lobe functions, emotional dysfunction, impaired pain empathy and disrupted functional connectivity of supplementary motor areas described in JME. In addition, the results support the theory of thalamocortial pathways being involved in the pathogenesis of JME.

Strong points:

The study provides more robust results regarding the volumetric changes in the gray matter of patients with JME compared healthy controls by including five more studies to the previously published study.

The study was done and the manuscript is written following PRISMA check list although the registration number is not provided.

 PRISMA flow diagram is provided properly regarding the study selection.

 Egger’s test and visual inspection of the funnel plots were done for checking the risk of bias.

 The replicability of the results was checked using Jacknife analysis.

 The heterogeneity analysis was done using random-effects model with Q statistics.

 Details of the included studies were mentioned in the result section except the reference 11.

The limitations of this study were adequately stated by the authors.

Weak point:

It is confusing that Figure 2B indicates decreased gray matter volume in the left insula although it was in the right insula in the manuscript (Line 137).

Minor

Please delete “from 2012” (page2, line 49) because three of five were published before 2012.

Montreal Neurological Institute of Talarirach space?

Spelling out abbreviations at the first appearance

e.g. MNI (page 2, line 56), SDM (page 2, line 60)

Consistency for using abbreviations

e.g. Juvenile myoclonic epilepsy JME (line 104), VBM (line 115) 

Author Response

Thank you very much for your kind comments. We now amended the text as per your suggestions, and highlighted the changed in red colour.

Reviewer 2 Report

Kazis et al. present a meta-analysis of voxel-based morphometry studies in juvenile myoclonic epilepsy.

The presented analysis, which included 12 papers published between 1990 and 2021, confirmed already known findings related to aberrant thalamo-cortical (frontal cortex) gray matter volumes.

My major concern is related to the methodology of meta-analysis. Meta-analysis is usually conducted in the context of a systematic review and the process of selecting studies, their critical appraisal, data extraction etc. should be clearly defined, which is not the case in the presented analysis. I also miss clear definitions of a research question and outcome measures. I would suggest consulting the prisma-checklist (please, see the link below) for presenting the methodology and results in line with this table.

http://prisma-statement.org/prismastatement/Checklist.aspx

Author Response

Dear Dr,.

Out study followed the prisma guidlines as clearly described in the methods section and as outlined by reviewer 1. We describe in details the findings of the studies, and we also evaluated the robustness of the results by visual inspection of the Funnel plots, and with Egger’s test.

Cao et al in their study included only 7 studies. In our study we included 12 studies in total.

Cao et al in their study had found increased gray-matter volume in the bilateral medial frontal gyrus and anterior cingulate cortex, and decreased gray-matter volume in the bilatelar thalamus.

In our study we found decreased gray-matter voume in the left thalamus, but in the right, and in the left insula.

We additionally found increased gray-matter volume in the left median cingulate/paracingulate gyri, the right superior frontal gyrus, the left precentral gyrus, the right supplementary motor area and the left supplementay motor area.

We do believe that our study corroborates some of the findings of Cao et al findings, and gives new insights to the volume changes in JME.

Best, The Authors.

Round 2

Reviewer 1 Report

This is the revised manuscript for “Gray matter changes in Juvenile Myoclonic Epilepsy. A Voxel-Wise Meta-Analysis”. The manuscript was adequately revised except for the minor points below in the abstract section and discussion.

“A decrease in the gray matter volume in the left thalamus, and in the right insula (line 21-22)” →in the left insula?

“Additional decreases of the volume of the gray matter were found in the left thalamus, and the right insula (line 201-202)”. → the left insula?

Author Response

Thank you! We did modify the 2 aspects you kindly suggested.

Reviewer 2 Report

I apologize if I missed that paper was designed according to prisma guidelines.

Author Response

Thank you!